# Deformable Medical Image Registration Using a Randomly-Initialized CNN as Regularization Prior

**Max-Heinrich Laves**   LAVES@IMES.UNI-HANNOVER.DE

**Sontje Ihler**   IHLER@IMES.UNI-HANNOVER.DE

**Tobias Ortmaier**   ORTMAIER@IMES.UNI-HANNOVER.DE

*Institute of Mechatronic Systems, Leibniz Universität Hannover, Germany*

## Abstract

We present deformable unsupervised medical image registration using a randomly-initialized deep convolutional neural network (CNN) as regularization prior. Conventional registration methods predict a transformation by minimizing dissimilarities between an image pair. The minimization is usually regularized with manually engineered priors, which limits the potential of the registration. By learning transformation priors from a large dataset, CNNs have achieved great success in deformable registration. However, learned methods are restricted to domain-specific data and the required amounts of medical data are difficult to obtain. Our approach uses the idea of deep image priors to combine convolutional networks with conventional registration methods based on manually engineered priors. The proposed method is applied to brain MRI scans. We show that our approach registers image pairs with state-of-the-art accuracy by providing dense, pixel-wise correspondence maps. It does not rely on prior training and is therefore not limited to a specific image domain.

**Keywords:** deformable registration, convolutional network, optical coherence tomography

## 1. Introduction

Deformable registration is a major challenge in medical image processing. The result is a dense mapping showing pixel-wise non-linear correspondences between a pair of images that best aligns the input image $I$ onto the target image $T$ by means of some similarity definition $\mathcal{L}$. Deformable registration is applied in the analysis of patient-specific temporal or anatomical changes, e.g. from pre-operative to post-operative state, or to show inter-patient variances (Sotiras et al., 2013). Deformable registration is also performed in atlas-based segmentation, where an input image is matched onto a target image with known segmentation (Cabezas et al., 2011).

Existing registration methods can be separated into two categories. The first category is based on non-learning methods which estimate a registration $w$ by optimizing a cost function of the form

$$\arg\min_{w} \left\{ \mathcal{L}(T, w \circ I) + \lambda \mathcal{R}(w) \right\} , \qquad (1)$$

where $w \circ I$ denotes $I$ warped by $w$. A common assumption of $w$ is a displacement or velocity vector field $\boldsymbol{u}(\boldsymbol{x})$. The final deformation results in $\boldsymbol{\phi}(\boldsymbol{x}) = \boldsymbol{x} + \boldsymbol{u}(\boldsymbol{x})$ which maps every pixel coordinate $\boldsymbol{x}$ to other pixel coordinates. The first term in (1) is referred to as data term, which is typically chosen to be a pixel intensity error measure. Optimization of the data

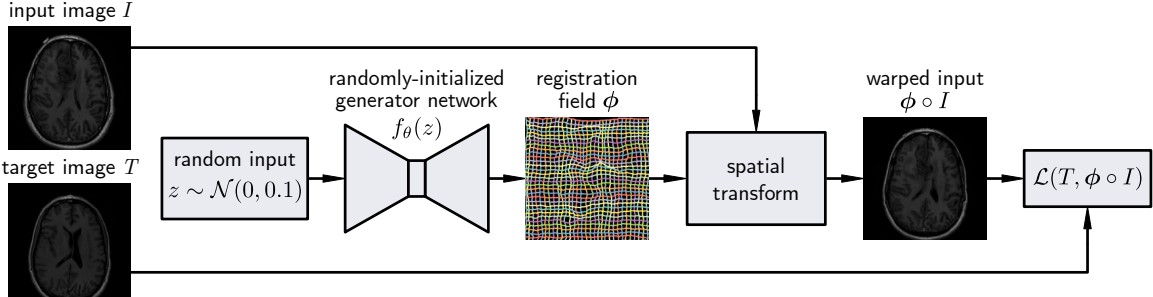

Figure 1: Overview of our method. The randomly-initialized generator network $f_\theta$ acts as parameterization of the registration field $\phi$. The parameters $\theta$ are optimized for every image pair individually by gradient descent.

term alone is considered ill-posed. The second term $\mathcal{R}$, weighted by trade-off factor $\lambda$, is a regularizer that shapes the registration by any chosen prior, which helps solving the ill-posed problem. Common regularization is done by enforcing smoothness onto the displacement vector field by penalizing first or higher order spatial derivatives of $\boldsymbol{u}$ (Werlberger et al., 2010). The result of the registration algorithm heavily depends on the cost function and therefore on the chosen prior of $\mathcal{R}$.

The second category implicitly learns the regularization prior by training a convolutional network on a large database of domain-specific images. Early approaches rely on ground truth registrations (Sokooti et al., 2017), which are hard to obtain especially in medical imaging. More recent methods (Balakrishnan et al., 2019) propose unsupervised registration using the spatial transformer function (Jaderberg et al., 2015). However, these methods either only support small displacements or require segmentation maps of the image pairs during training to assist the convergence (Hu et al., 2018). Additionally, the trained networks are limited to register images from the training domain (e.g. CT or MRI).

Inspired by the idea of deep image priors (Lempitsky et al., 2018), we subsequently propose our learning-free method for deformable medical image registration using the structure of an untrained convolutional network as regularization prior.

## 2. Methods

Lempitsky et al. have recently shown that excellent performance of CNNs for inverse image problems, such as denoising, is not only based on their ability to learn image priors from data, but is also based on the structure of a convolutional image generator itself (Lempitsky et al., 2018). They gave evidence that the structure of a network alone is sufficient to capture enough image statistics to provide state-of-the-art performance in inverse image tasks.

Leveraged by this idea, we reformulate the task of deformable image registration by using the structure of a convolutional network as regularizer (see Fig. 1). An image generator network $\boldsymbol{u} = f_\theta(z)$ with randomly-initialized parameters $\theta$ is interpreted as parameterization of the dense displacement field $\boldsymbol{u} \in \mathbb{R}^{2 \times H \times W}$ from which the deformation $\boldsymbol{\phi} = \boldsymbol{x} + \boldsymbol{u}$ between an input image $I \in \mathbb{R}^{C \times H \times W}$ and a target image $T \in \mathbb{R}^{C \times H \times W}$ can be obtained by adding to

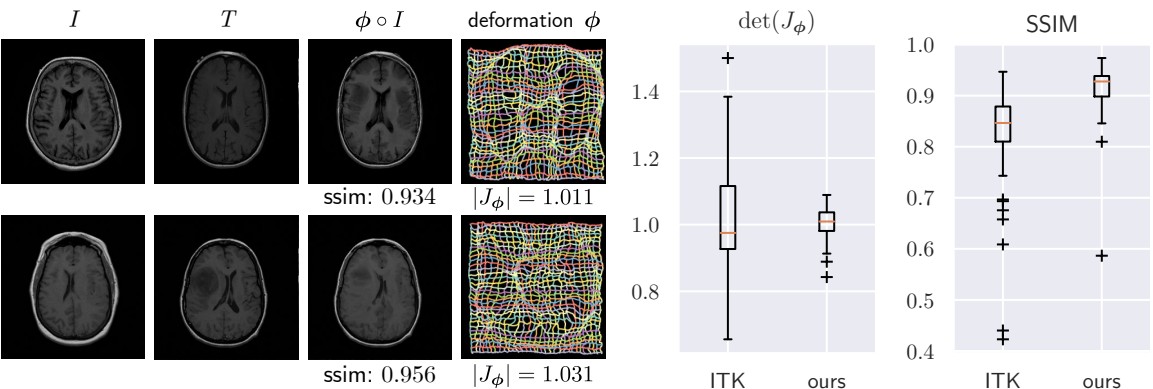

Figure 2: Results of our approach compared to a state-of-the-art method from the Insight ToolKit (ITK). Left: Two example MRI pairs from the data set. Right: Boxplots of the means of $\det(J_\phi)$ and SSIM between $T$ and $\phi \circ I$.

the identity warp $\boldsymbol{x}$. The input $z \in \mathbb{R}^{C' \times H \times W} \sim \mathcal{N}(0, 0.1)$ has the same spatial dimensions as $\phi$ and is sampled from a random normal distribution in every iteration. This leads to the following optimization problem

$$\arg\min_\theta \left\{ \mathcal{L}\left(T, (\boldsymbol{x} + f_\theta(z)) \circ I\right) \right\}, \tag{2}$$

where $(\boldsymbol{x} + f_\theta(z)) \circ I$ denotes the differentiable spatial transformer function (Jaderberg et al., 2015). Eq. (2) is optimized for every image pair $\{I, T\}$ using the Adam gradient descent optimizer (Kingma and Ba, 2014). As data term, we chose pixel-wise mean absolute error $\mathcal{L}(T, \phi \circ I) = |(\phi \circ I) - T|$. The architecture of the image generator network $f_\theta$ is chosen according to (Lempitsky et al., 2018). It has an encoder-decoder structure with skip connections between the encoding and decoding part. To begin the optimization from close to an identity warp, we initialize the parameters with $\theta \sim \mathcal{N}(0, 0.01)$.

## 3. Results & Conclusion

We demonstrate our approach on the task of 2D brain magnetic resonance imaging (MRI) registration. The data used in this work contain 109 pairs of MRI scans from The Cancer Genome Atlas (NCI, 2019) showing lower-grade gliomas. We use the structural similarity index (SSIM) (Wang et al., 2004) between $\phi \circ I$ and $T$ and the mean of the determinants of Jacobians $\det(J_\phi)$ (Ashburner, 2007) of the deformation as evaluation metrics. The latter metric shows regularity of $\phi$. We compare our method to state-of-the-art methods from the Insight ToolKit (ITK) registration framework by combining an initial affine registration and a subsequent deformable displacement field registration (Avants et al., 2012). Results for exemplary image pairs and boxplots of results for all image pairs are shown in Fig. 2. Additional results including registration fields are shown in appendix A.

The results reveal that the structure of a convolutional network can act as regularization in deformable medical image registration with state-of-the-art performance. This connects traditional non-learning methods and learning-based methods by using randomly-initialized convolutional networks as prior.

## Acknowledgments

This research has received funding from the European Union as being part of the EFRE OPhonLas project.

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

## Appendix A. Results

|      | SSIM            | $\det(J_\phi)$  |
| ---- | --------------- | --------------- |
| ITK  | $0.827 \pm 0.096$ | $1.036 \pm 0.254$ |
| ours | $0.913 \pm 0.051$ | $1.007 \pm 0.045$ |

Table 1: Mean results of SSIM between $T$ and $\phi \circ I$, and determinants of Jacobian $\det(J_\phi)$ of deformation $\phi$ compared to a state-of-the-art method from ITK.

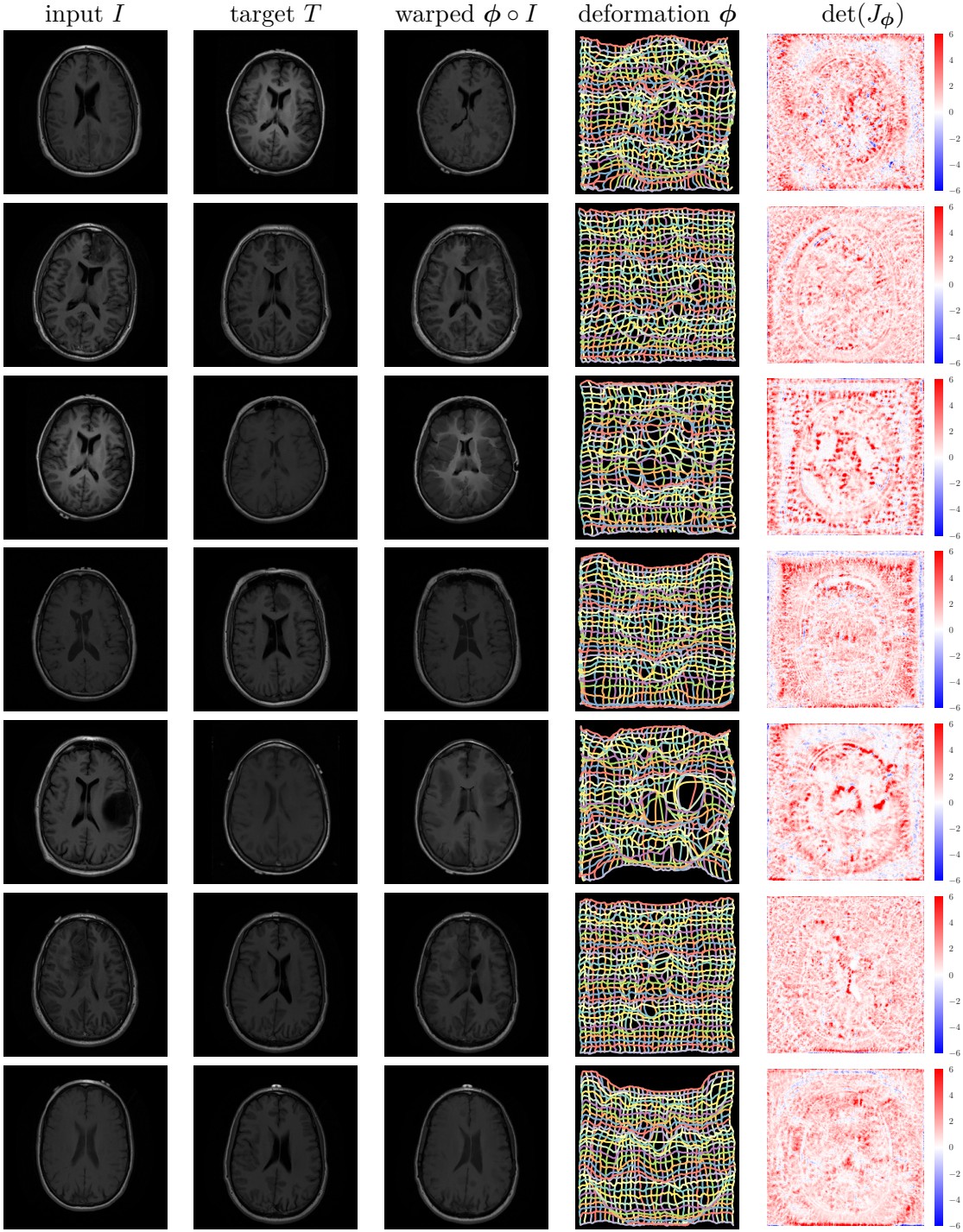

Figure 3: Additional results showing image pairs $\{I, T\}$, warped input $\phi \circ I$, estimated deformation grid $\phi$ and map of determinants of the Jacobian matrix $J_\phi = \nabla\phi$ for every entry of $\phi$. $J_\phi$ shows local regularity of the deformation field. The deformation is diffeomorphic, where $\det(J_\phi) > 0$.

