# OpenReview forum: "Deformable Medical Image Registration Using a Randomly-Initialized CNN as Regularization Prior"
_MIDL.io/2019/Conference/Abstract — MIDL Abstract 2019_

### Official Review · AnonReviewer1 · 2019-04-28
**original contribution on challenging problem - improve assessement of the deformation smoothness**

**Rating:** 4
**Confidence:** 3

**Review:**

Deformable Medical Image Registration using a randomly initialized CNN as regularization prior

The paper proposes to exploit deep image priors in the learning process of deformable registrations. The method uses recent advances in prior generations (Lempitsky+Jaderberg) to optimize for 2D image registration via random-sampled generated deformations. The results look to indicate a superiority in similarity scores over a conventional method (Avants).
The paper shows promising results with an original methodology. To further strengthen the contribution, the regularity of the registration maps should be evaluated. This is typically done by shows the determinant of Jacobian maps of the deformation field (how distorted is the map). Results should also use conventional evaluation scores (Dice, Jacobian maps, Hausdorff) to fully appreciate the superiority or inferiority of the proposed method.
Ref on Kingma, NeuRIPS instead of arxiv

Best abstract of reviewed batch of 8.

---

### Official Review · AnonReviewer2 · 2019-05-01
**Potentially interesting; but very speculative with no attempt to provide evidence for the main claim of built-in regularity**

**Rating:** 2
**Confidence:** 3

**Review:**

The work looks into pairwise image registration. The abstract proposes a regularization-free formulation with a deep learning parametrization of deformations inspired by Deep Image Priors (Lempitsky et al).

The parametrization is a mapping f_theta(z), where z is a white noise input image and theta are the free parameters in a convolutional architecture f (encoder-decoder architecture with skip connections). z is resampled every iteration(?) during the optimization.

| Strengths:
- Originality, as a formulation of registration.
- The paper is clear.

| Weaknesses:
- No evidence to support the main claim of the authors, that is to say: that regularization in the cost function is unnecessary because it is a feature of the parametrization. It is also far from obvious looking at the formulation.

- If only one result, the smoothness of the estimated displacements should be assessed, rather than the data similarity term.

- The caption is missing in Fig. 3, so the colour rendering of the flow is open to interpretation. Does it encode direction and magnitude of the displacements? The scale is also missing. A more straightforward rendering as a vector field, a displacement grid, or displaying a scalar map showing some statistics on the smoothness of the displacements would be better.

Judging from reported triplets of (source, target, warped) images, it is doubtful that the estimated displacements are regular. A few examples:
- Fig. 3, last two rows. The source images have wiggly patterns but neither the target nor the warped images do. If the displacements were smooth, the warped images should retain wiggly patterns.
- In the 3rd row and the 1st to last, there is no uniformly black area in the source images but there is such an area in the warped images (for this reason they better match their respective target images in terms of intensity). Again, this should not happen with diffeomorphic or even continuous displacements.
- First row, top part of the target image, there is a white wiggly line extending from both extremities and vanishing near the center. There is no such pattern in the upper half of the source image, but there is a similar pattern in the warped image. Again same conclusion.

- How to interpret the kind of "supervoxel" structures with sharp discontinuities that are noticeable in the warped image upon zooming in? (same in the flow images)

---

### Decision · Program_Chairs · 2019-05-06
**Acceptance Decision**

Accept